# Deep Generative Inpainting with Comparative Sample Augmentation

## Abstract

Recent advancements in deep learning techniques such as Convolutional Neural Networks(CNN) and Generative Adversarial Networks(GAN) have achieved breakthroughs in the problem of semantic image inpainting, the task of reconstructing missing pixels in given images. While much more effective than conventional approaches, deep learning models require large datasets and great computational resources for training, and inpainting quality varies considerably when training data vary in size and diversity. To address these problems, we present in this paper a inpainting strategy of *Comparative Sample Augmentation*, which enhances the quality of training set by filtering out irrelevant images and constructing additional images using information about the surrounding regions of the images to be inpainted. Experiments on multiple datasets demonstrate that our method extends the applicability of deep inpainting models to training sets with varying sizes, while maintaining inpainting quality as measured by qualitative and quantitative metrics for a large class of deep models, with little need for model-specific consideration.

## 1 Introduction

Semantic image inpainting, the task of reconstructing missing pixels in images, has various applications in computer vision problems such as computational photography and image restoration (Yu et al. (2018)). Although there has been substantial progress in relevant research, image inpainting still remains a great challenge due to the difficulty to synthesize missing pixels that are visually and semantically coherent with surrounding existing background pixels. Such an issue becomes especially apparent when the amount of available training image data is limited due to the current limitations of deep models in representing possible latent features.

Current solutions to the inpainting problem mainly belong in two groups: traditional patch-based learning methods and deep learning methods. Traditional methods often directly utilize background information by assuming that information of missing patches can be found in background regions(Barnes et al. (2009)), leading to poor performances in reconstructing complex non-repeating structures in the inpainting areas and in capturing high-level semantics. Deep learning neural methods, on the other hand, exploit deep models to extract representations of latent space of existing pixels and transform inpainting into a conditional pixel generation problem(van den Oord et al. (2016), Yu et al. (2018)). While these approaches did produce images of significantly higher quality, they generally require an enormous amount of highly varied training data for model training, a requirement of which makes it impossible to apply these strategies when the set of available training data is limited. Recent research (Goodfellow et al. (2015)) also suggest that the performances of neural networks vary considerably in a variety of tasks when input images contain adversarial noise that potentially affects latent space, showing that countering adversarial examples is a key in boosting deep learning models.

To address these issues, we propose in this paper a simple black-box strategy that easily adapts to existing generative frameworks without model specific considerations and extends deep neural network strategies to the cases with varying amounts of training data. The contribution of this paper can be summarized as follows:

- We designed an effective strategy of selecting relevant samples by constructing a similarity measure based on color attributes of the inpainting image and the training dataset and selecting the K-most-relevant pictures.

- Our algorithm also bolsters local image qualities by adding new images created through white-noise addition on the original inpainting image.

- We have also conducted detailed set of experiments using the state-of-the-art generative inpainting structures, and demonstrate that our method achieves better inpainting results when the available training data is not necessarily abundant.

## 2 RELATED WORK

Prior work on image inpainting can be classified into two groups: ones that either directly incorporate features to be learned within the image to be inpainted, and ones that utilize learning methods in relation to other possibly available training images to extract representation of latent space. Traditional approaches usually involve algorithms to directly handle information from background to the missing pixels(Yu et al. (2018)). These methods borrow surrounding textures very nicely and achieve satisfactory performances when the inpainting area contains mostly repetitive simple features, but fail at more complex inpainting images such as human faces and natural scenery. Additionally, methods which extensively handles patch similarity(Simakov et al. (2008)) tend to be computationally expensive, making the method less applicable for cases where the training data and computational resources are both limited.

On the other hand, deep learning based methods using Convolutional Neural Networks and Generative Adversarial Networks encode high-level and low-level features via an encoder-decoder network, and proceed by using constructing objective optimiziers which take consistency factors into consideration(Iizuka et al. (2017)). Such design indeed enables the model to generate more diverse content in structured images, but effective training for satisfactory performances of these models require access to huge amounts of varied labeled data often unavailable in real-time applications. Moreover, as observed in Yu et al. (2018) and Iizuka et al. (2017), training generative inpainting models requires high computational resources and training time up to weeks and months due to the high complexity of the current network structures.

## 3 SAMPLE AUGMENTATION

In this section we propose our data augmentation method of Comparative Sample Augmentation, which consists of two separate parts: a comparative augmenting filter which selects the most relevant samples from the training dataset, and a self enricher which adds noise masks to the original image to produce additional images for training purposes. Images chosen by these two procedures are then combined together to form the dataset for subsequent inpainting neural network training. The first part of our algorithm takes global information about the training set into consideration, while the second part tackles the local features within the image to be inpainted. All the steps in our algorithm are easily to implement and adaptable to a variety of generative adversarial models.

### 3.1 COMPARATIVE AUGMENTING FILTER

One notable problem in many tradition inpainting methods(Barnes et al. (2009),Simakov et al. (2008)) is the under-representation and even neglect of possible contextual information present in other similar images. Prior deep learning methods (Iizuka et al. (2017), Yu et al. (2018)) background pixels into consideration, but the training time overhead in magnitudes of weeks/months makes scalable adaptations hard and even infeasible.

To counter such problems, we introduce a comparative augmenting filter on the training dataset before inpainting. Motivated by the $l_1$ reconstruction loss as described in Yu et al. (2018), we first define a similarity distance between two discrete distributions $P$ and $Q$ as below:

$$d(P,Q) = \frac{\|P - Q\|_1}{2\max(\|P\|_1, \|Q\|_1)}$$

This particular distance normalizes to [0,1] and becomes 0 if and only if $P = Q$ almost surely and is relatively easy to compute. For our purposes, we consider the distributions of RGB pixels in the two images, where values of red, green, blue range from 0 to 255. Given the training dataset and the image to be inpainted, we compute the distances between the color distribution of our image and those of other pictures, and select the $K$-closest images with respect to the distance. This strategy collects the most relevent images as measured by color, and helps inpainting by filtering out other irrelevant images whose contrasting color may adversely effect inpainting results when the inpainting model uses global information to learn possible choices of missing images.

## 3.2 SELF ENRICHMENT

In addition to considerations on global information and consistency, we also bolster the robustness of the deep inpainting model by adding random masks onto the image to be inpainted. Our motivation is that the deep neural networks, due to the over-dependency on the training data, may omit other possibly useful latent features during training and yield unsatisfactory output for the given tasks when the input contains advarsarial noise. Goodfellow et al. (2015), for instance, noted that neural networks may assign incorrect labels to images in classification tasks possibly due to the inherent limitations of the training dataset. Since most of the generative adversarial networks are implemented with deep neural network, the narrow representation power of deep networks may carry over to insufficient latent space representation which ultimately results in unsatisfactory generated images and fluctuating performances due to potential overfitting.

In view of such concerns, we consider further enriching the training set with additional images constructed by additional normal random masks. Given the inpainting image, we add a batch of random normal noise to each background pixel so as to produce varied noised images that are similar to the original image. This batch of images is to be included in the training data to place emphasis on the contextual clues specific to the image itself. Together with the training images that reflect information about possible global similarity, the local images thus selected complement the training outcome by providing information with robustness.

## 4 EXPERIMENTS

As part of our experiments, we have evaluated our method on the datasets: CIFAR-10, CelebA, and Places. To test the applicability of our method to cases of "many" and "few" pieces of data, we randomly sample 5000 images over the datasets Cifar-10 For the case with "few" pieces of data, we select the "Ocean","Orchard","Pier" subsets from the Places dataset. In both cases, we randomly select an image as the inpainting target by masking it with an black square hole in center. These reduced datasets are named in our subsequent experiments Reduced-CIFAR, Reduced-CelebA, and Reduced-Places respectively.

By the observations in Gulrajani et al. (2017) and Yu et al. (2018), we utilize the state-of-the-art WGAN-GP for our inpainting generative model framework. Notice that the procedures in our algorithm does not rely on the specific details of generative models, and that our strategy can easily be included in most current inpainting frameworks.

### 4.1 QUALITATIVE EVALUATIONS

For demonstration, figure 1,2 below are a couple of generated images by the GAN with comparactive sample augmentation and those by GAN without. We observe in our experiment that the GAN with our sample augmentation produces better or on-par images no later than the original GAN structure does.

### 4.2 QUANTITATIVE EVALUATION

We also notice that image inpainting lacks effective quantitative evaluation metrics, so we follow the usual practice(Iizuka et al. (2017),Yu et al. (2018)) to report the mean $l_1$ and $l_2$ errors of our reconstruction. Due to time constraints, we for now only include our result for Reduced- Cifar-10, and will report that in more complete version.

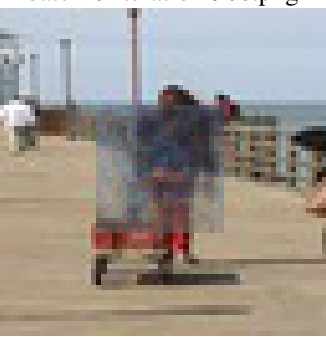

workshop format/impainted-image-batch-0-iteration-900.png

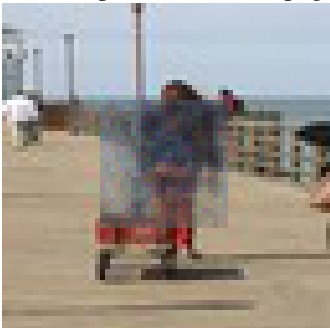

workshop format/result-2.png

Figure 2: Inpainting without augmentation

Figure 1: The image to be inpainted

| Datasets | Augmentation | No Augmentation |
|----------|--------------|-----------------|
| Reduced-CIFAR | 16.9 percent | 17.4 percent |

Table 1: $l_1$ and $l_2$ errors of inpainting with respect to datasets

## 5 CONCLUSION AND FUTURE WORK

In sum, we introduce in this paper the strategy of comparative sample augmentation for deep inpainting. We show by experiments that our method extends the deep-learning-based inpainting methods to the cases with varying sizes of numbers of data, without any need for model specific adjustments.

As a part of our future work, we plan to add additional strategies such as gradient matching and feature encoders into the 1st part of our algorithm to better approximate the distance between the inpainting image and the training images. Further directions include better control of the augmentation in the 2nd part of our method using recent advancements such as Cubuk et al. (2018) in data augmentation.

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
