# OpenReview forum: "Deep Generative Inpainting with Comparative Sample Augmentation"
_ICLR.cc/2019/Workshop/LLD — Submitted to LLD 2019_

### Official Review · AnonReviewer2 · 2019-04-06
**Not clear contributions; not even clear method**

**Rating:** 2
**Confidence:** 2

**Review:**

The authors propose an augmentation method for image inpainting; the core idea lies in filtering out irrelevant images and augmenting the images by adding random normal noise in the pixel space.

Even though potentially, the method has its merits, the method is not clearly written, while the paper is rather confusing (please see below).

Major points:

1) There is no formal definition of the task in the paper (i.e. they interchangeably just mention inpainting or semantic inpainting without any definition). In conjunction with the lack of clear explanation of their method, understanding the contributions is challenging.

2) It is not clear what type of GAN the authors use. They mention WGAN-GP (Gulrajani et al), but it is not mentioned whether they generate from scratch the images or whether this is a conditional GAN (as traditional inpainting methods). Therefore, it is not clear how GAN is used in this work.

3) The comparative augmentation filter seems like a KNN in the pixel-space; given question 2, it is not clear where this filter fits in the training method.

4) The authors mention that their method 'extends the applicability of deep inpainting methods', but in the end only experiment in a 'restricted-CIFAR'; the rest experiments of Celeb-A and Places are not included.

5) The propose augmentation method, i.e. adding random normal noise per pixel is not novel; in addition, there is no ablation experiment demonstrating the benefits experimentally.

6) Why do the authors propose to filter the images in the pixel space and not a perceptual space as popular the last few years?


Minor points:
1) The following expressions should be more rigorous:
   - 'becomes 0 if and only if P = Q almost surely'.
   - 'produces better or on-par images no later than the original GAN'.
   - 'and ones that utilize [...] latent space.'

2) The authors mention that Yu et al 'fail at more complex inpainting images such as faces and natural scenery'. Do they have some visual examples that contradict the original paper? Because in the original paper, the methods perform well in both domains.

3) In Fig. 1 and 2, what is the '/workshop_format/[...].png'?

4) In table 1, l1 and l2 errors are mentioned, but there is a single number; which also includes an undefined percentage. Could the authors clarify what they mean?

5) The authors do not mention how much they augment their original images, i.e. for every original image how many images are used during training? Is the noise sampled per image or per batch?

Given the major improvement points; the paper should be re-written before being accepted to the workshop.

---

### Official Review · AnonReviewer1 · 2019-04-07
**Requires better justification, analysis, and experiments**

**Rating:** 1
**Confidence:** 3

**Review:**

Summary:

Help image inpainting using GANs by two strategies:

1) Comparative Augmenting filter: choose images from a training dataset whose histogram is similar to the image in consideration. Histogram matching is an old trick in the computer vision community.

2) Self-Enrichment: add random noise to each pixel. This seems to be the same as "Instance Noise" [1], which the authors did not cite but claim as their own.

The authors motivate their strategy by saying older methods don't work well in case of non-repetitive backgrounds such as faces, but themselves rely on a global similarity like histogram matching. Highly doubt if this can theoretically work.  Results in the paper show that practically the improvement is negligible.

Authors mention 3 contributions but do not justify their claims:
1) Histogram matching - authors don't mention that it is an old trick, and don't justify why this could work. Also, practical results show that it doesn't.
2) Instance noise - authors don't cite an older paper that proposed the same. Also, practical results show that it doesn't.
3) Authors mention "detailed set of experiments" in the introduction but only include 1 in the experiment section, and say they could not add more due to time constraints.

Literary errors:
There are quite a few word-level errors such as word redundancy, sentence errors, spelling mistakes that make the paper difficult to read.

[1]  "Instance Noise: A trick for stabilising GAN training" https://arxiv.org/abs/1610.04490

---

### Decision · Program_Chairs · 2019-04-08
**Acceptance Decision**

Reject